



# Characterization and Mechanical Testing of Manufacturing Defects Common to Composite Wind Turbine Blades

Jared W. Nelson[1], Trey W. Riddle[2], and Douglas S. Cairns[3]

[1]SUNY New Paltz, Division of Engineering Programs, New Paltz, NY
[2]Sunstrand, LLC, Louisville, KY
[3]Montana State University, Dept. of Mechanical and Industrial Engineering, Bozeman, MT

**Abstract**. The Montana State University Composites Group performed a study to ascertain the effects of defects that often result from the manufacture of composite wind turbine blades. The first step in this multi-year study was to
systematically quantify and database these defects before embedding similar defects into manufactured coupons. Through the Blade Reliability Collaborative, it was determined that the key defects to investigate were fiber waves and porosity. An inspection of failed commercial-scale wind turbine blades yielded metrics that utilize specific parameters to physically characterize a defect. Methods to easily and consistently discretize, measure, and assess these defects based on the identified parameters were established to allow for statistical analysis. Data relating flaw
parameters to frequencies of occurrence were analyzed and found to fit within standard distributions. Additionally, mechanical testing of coupons with flaws based on this physical characterization data was performed to understand effects of these defects. Representative blade materials and manufacturing methods were utilized and both material properties and damage progression were measured. It was observed that flaw parameters directly affected the mechanical response. While the data gathered in this first step is widely useful, it was also intended for use as a
foundation for the rest of the study; to perform probabilistic analysis and comparative analysis of progressive damage models.

## 1 Introduction & Background

With the rapid growth of the wind segment of the energy market, it is important that wind farms achieve maximum availability by reducing down time due to maintenance and failures. Since most components of a wind turbine are
located over 60 m (200ft) above the ground and are large and complex, performing repairs on site is costly and difficult. Repairs to these systems not only require the turbine to be shut down but the problematic system will often require a crane for removal. This is especially true for wind turbine blades where advanced composite materials have become an optimal choice due to their high strength-to-weight ratio. Lower cost composite materials and manufacturing methods have become the standard for wind turbine blades to compete with traditional energy production technologies.
While the resulting final cost can be up to two orders of magnitude less than a typical aerospace composite structure, the inclusion of manufacturing defects is more likely. It has been inferred through analysis of wind turbine down time due to blade failures resulting from such manufacturing defects, that design and manufacturing within the wind industry does not always ensure a 20-year design life (Hill et al., 2009). A comprehensive study to characterize and understand the manufacturing flaws common in blades has not been performed within the public domain; however,



research has been performed to better understand what is needed to improve blade reliability (Hill et al., 2009; Red, 2008; Walford, 2008; Veldkamp, 2008). The Department of Energy sponsored, Sandia National Laboratory led, Blade Reliability Collaborative (BRC) was formed in large part to address this issue. The research described herein compiles the first stage of a multi-year program performed by Montana State University (MSU) within two areas—Flaw

Characterization and Effects of Defects.

The primary goal of the MSU research initiative has been to develop a protocol which can be employed in a quality assurance and reliability program to quantify the implications of wind turbine blades containing manufacturing defects to ensure blade life while reducing costly repairs. In turn, these methods may then be used to improve blade manufacturing and design procedures. The function of the Flaw Characterization portion of this program has been to

provide quantitative analysis for two major directives: acquisition and generation of quantitative flaw data describing common defects in composite wind turbine blades; and, development of a flaw severity designation system and probabilistic risk management protocol for as-built flawed structures (Riddle et al., 2011). The Effects of Defects portion focused on the development of state-of-the-art modeling capabilities, correlated to experimental data, to predict the mechanical response of included flaws (Nelson et al., 2011). As such, a foundational work to characterize

typical defects and ascertain their effects on mechanical performance was performed.

## 2 Wind Industry Blade Survey and Flaw Characterization

The Blade Reliability Collaborative (BRC) directed the MSU team to investigate the effects of porosity, in-plane (IP) and out-of-plane (OP) waves shown in Figure 1 and Figure 2, respectively. Based on statistical commonality in wind turbine blades, it was critical to the development of this program to identify the precise geometric nature of these

flaws. To do this, several commercial scale wind turbine blades were reviewed. This data set, while relatively small, provided a strong starting point for the entire project and specifically for the development of a protocol by which other blades may be examined and flaws may be characterized going forward. These techniques are not manufacturer or size dependent, and furthermore, could be applied to any composite structure in theory.

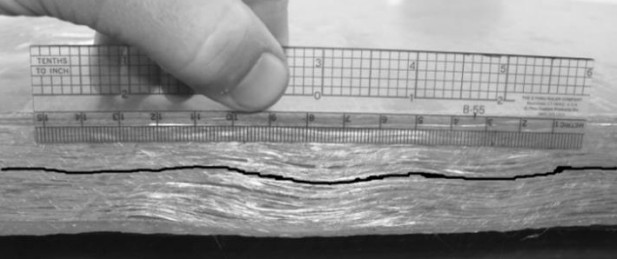

**Figure 1: Example of an out-of-plane (OP) wave where fiber waviness is observed through the thickness.**




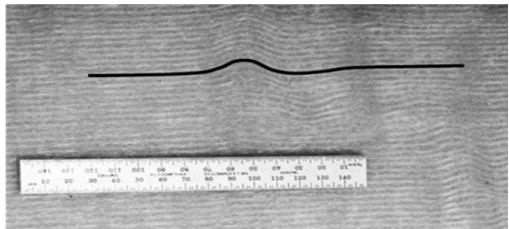

**Figure 2: Example of an in-plane (IP) wave where fiber waviness is observed in-plane with the outer surface.**

The process by which in-plane and out-of-plane wave data were collected was essentially the same. An image processing software was used on photographs of as-built flawed blade sections where each defect feature was manually

traced with a line as shown in black in both **Error! Reference source not found.** and **Error! Reference source not found.**. A separate processing script was written to extract the spatial coordinate data of the traced defect line. From these data, each complete wave form was discretized into separate individual waveforms. One example of a complete wave and the waveform discretization process is shown in **Error! Reference source not found.**.

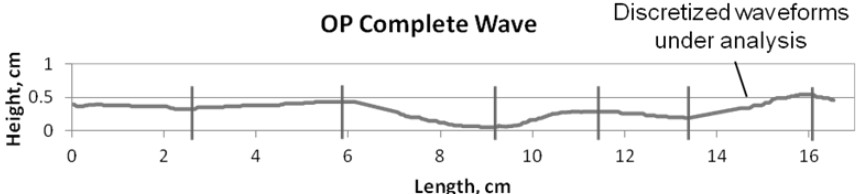

**Figure 3: Example of OP waviness, complete spatial data, and descretization positions.**

Each discretized wave form's geometry was then mathematically characterized; cubic spline (Equ 1) and sinusoidal curve (Equ 2) fits were both evaluated for their applicability to mathematically describe the wave perturbation:

$$Y = Ax^3 + Bx^2 + Cx + D \tag{1}$$

$$Y = E + F \sin\left(\frac{2\pi}{\omega x + \varphi}\right) \tag{2}$$

where $A$, $B$, $C$, and $D$ are polynomial coefficients, $E$ is the offset, $F$ is the amplitude, $\omega$ is the wavelength, and $\varphi$ is the phase.

In order to optimize the goodness-of-fit of the wave spatial data to the mathematical formulations, a user built least squares regression algorithm was used. This function utilizes the Generalized Reduced Gradient (GRG) constrained optimization algorithm. In a least-squares data fitting method, the most accurate model is established by minimization of the sum of squared residuals. A residual being the difference between an observed value and the fitted value provided by a mathematical model. The GRG algorithm was used to manipulate model values ($A$, $B$, $C$, $D$, $E$, $F$, $\omega$, $\varphi$)

until the sum of the squares was minimized (Biegler, 2011).

Both models, using spline and sinusoidal fits, yielded similar goodness-of-fit tendencies. The sinusoidal analysis proved to be faster and was utilized on bulk data analysis. Moreover, the ability to reference model parameters, which





have a direct physical meaning (e.g. amplitude and wavelength), was useful in performing statistical characterization of wave parameters. Once a fit was performed, each wave segment was characterized in terms of wavelength, amplitude and off-axis fiber angle (**Error! Reference source not found.**). While previous studies have used aspect ratio or wave severity (amplitude/wavelength) instead of fiber angle as a metric for characterization, such

quantification may be slightly more challenging in the field since aspect ratio requires knowing both the amplitude and wavelength (Adams and Hyer, 1993; Adams and Bell, 1995; Mandell et al., 2003). Even though it is possible that only the fiber angle can be measured directly in the field, wave amplitude ($A$), wavelength ($\lambda$), and off-axis fiber angle ($\theta$) were characterized, as shown in **Error! Reference source not found.**, for comparative purposes.

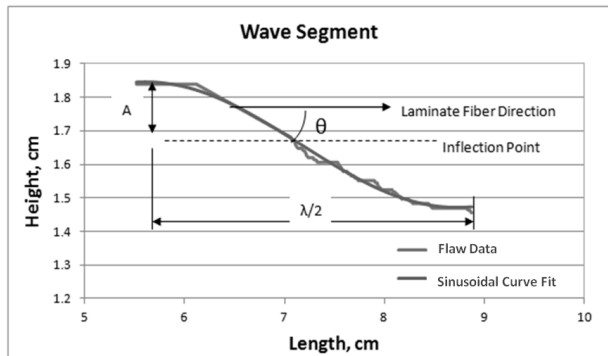

**Figure 4: A sine wave superposed on a segment of the OP waviness shown in** Error! Reference source not found.**.**

### 2.1 Determination of Defect Parameters for Testing

Characterization of the various wave flaws found in the field data yielded 63 OP and 48 IP independent, discrete waveforms. Values for amplitude and wavelength of each instance are shown in **Error! Reference source not found.** where it may be seen that there is significant variability within the data. However, the data are well grouped, indicating

consistency in the manufacturing processes. The resulting off-axis fiber angles from these OP and IP waves are shown in **Error! Reference source not found.**. Specific attention should be paid to the outlying group of angles highlighted by the circle in left of **Error! Reference source not found.**. The reader should note that these angles were collected from blade sections which failed at these out-of-plane flaw locations, and therefore, these magnitudes likely include plastic deformation.

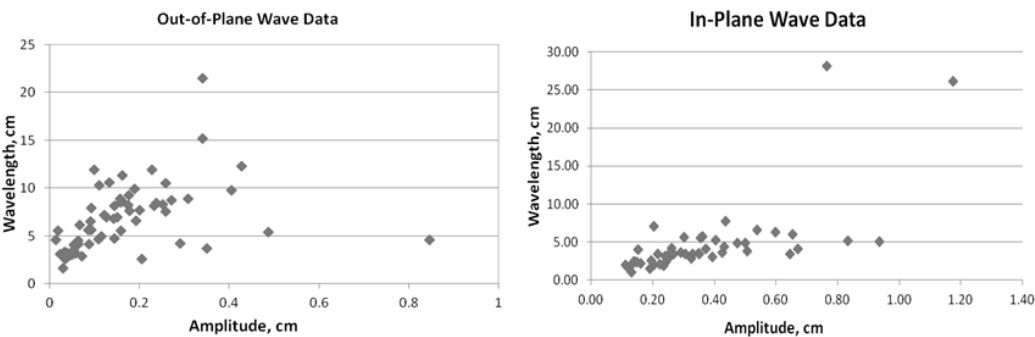




**Figure 5: Collected OP and IP wave data (left-to-right, respectively).**

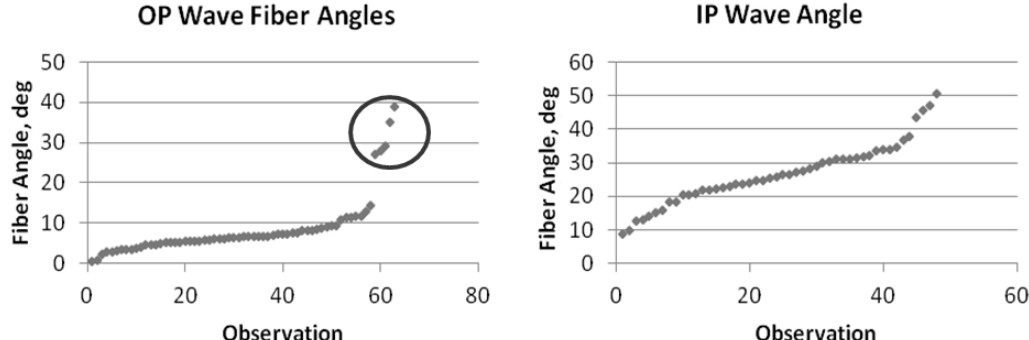

**Figure 6: OP and IP wave off axis fiber angles from data shown in** Error! Reference source not found. **(left-right, respectively).**
Mean and standard deviation values were used to develop Normal distributions to describe the frequency of flaw
magnitude occurrences and Weibull distributions (2 parameter) were generated using Maximum Likelihood
Estimation. To develop frequency of occurrences distributions, fiber angle values were binned together into groups as
shown in **Error! Reference source not found.**. For OP waves, angles were binned in one-degree increments while
for IP waves, angles were binned in four-degree increments. The frequency of each fiber angle can be seen in **Error!
Reference source not found.** where the observed frequency of occurrence is displayed with applicable Weibull and
Normal distribution curves. In general, wave fiber angle values show a strong inclination towards common
distributions such as the Weibull and Normal distributions for both cases. Similar binning procedures where applied
to amplitude and wavelength data for both wave types; however, the distributions were less accurate further justifying
characterization with fiber angle. Generalized information regarding both IP and OP wave group data is summarized
in **Error! Reference source not found.**.

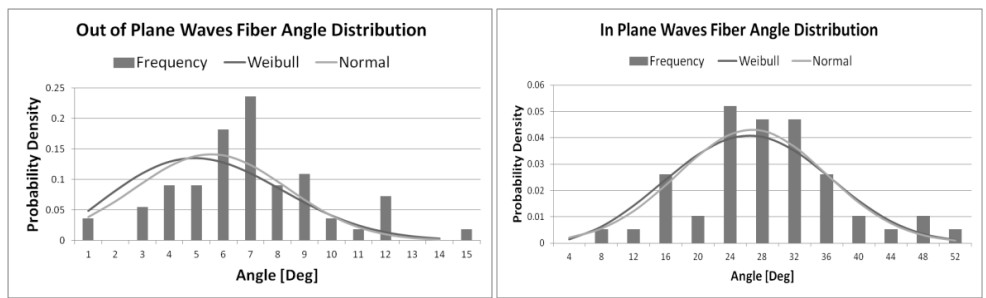

**Figure 7: OP and IP wave fiber angle distributions (left-to-right, respectively).**



**Table 1: Summary of wave data generated from processes outline above.**

| OP Waves | Amplitude, cm | Wavelength, cm | Fiber Angle, deg |
|---|---|---|---|
| Min | 0.02 | 1.58 | 0.59 |
| Max | 0.85 | 21.49 | 39.01 |
| Mean | 0.17 | 6.74 | 8.55 |
| Standard Deviation | 0.11 | 3.00 | 2.82 |
| **IP Waves** | **Amplitude, cm** | **Wavelength, cm** | **Fiber Angle, deg** |
| Min | 0.11 | 1.08 | 8.68 |
| Max | 1.18 | 28.12 | 50.66 |
| Mean | 0.37 | 4.75 | 26.73 |
| Standard Deviation | 0.23 | 4.96 | 9.26 |

A test program focused on the characterization of the mechanical performance of specimen with scaled flaws was developed from the data collected in this study. Given the scale difference between the blades and coupon-sized test

specimen, the flaws were scaled using a Weibull scaling analysis where fracture strength was adjusted with material volume. Based on this "weakest link" theory, as material volume decreases, the population of defects also decreases, thereby reducing the probability of a failure from a flaw. The ratio of fracture strengths may then be found if the probability of survival is assumed to be the same for both small and large-scale composite structures:

$$\frac{\sigma_1}{\sigma_2} = \left(\frac{V_2}{V_1}\right)^{\frac{1}{m}} \tag{3}$$

where $\sigma_{1,2}$ are the fracture strengths, $V_{1,2}$ are the volumes and $m$ is the Weibull modulus.

Using this method of scaling, three wave forms for each type of flaw were systematically chosen, as shown in **Error! Reference source not found.**, for testing based on geometry characterization and statistical significance while representing data points around an angular region of interest. The parameters for waves OP1 and OP2A were identified to be included due to the similarity in fiber angle occurring from different a different combination of amplitude and wavelength. The additional OP wave (OP4A) had mean values for all three parameters, and therefore, landed in the

center of all the parameter distributions. As such, these data points combined to sufficiently described an OP wave common to the specific wind turbine application. By design, the mean value also delivered baseline values for comparison of the effects of amplitude and wavelength independently with the OP1 and OP2A results. In-plane test waves IP2 and IP3 followed this same approach as the OP where they each had different amplitudes and wavelengths, but resulted in similar misalignment angles. Similarly, the IP1 case represented the parametric mean for all values.

**Table 2: The OP and IP wave parameters used in the coupon testing program.**



| OP1 Wave [mm] | As-Built | Scaled |
|---|---|---|
| Max Amplitude | 8.5 | 2.9 |
| Mean Wavelength | 67.4 | 22.8 |
| Angle [deg] | 34.9 | 36.8 |

| IP1 Wave [mm] | As-Built | Scaled |
|---|---|---|
| Mean Amplitude | 3.7 | 1.9 |
| Mean Wavelength | 47.5 | 23.8 |
| Angle [deg] | 24.8 | 24.8 |

| OP2A Wave [mm] | As-Built | Scaled |
|---|---|---|
| Mean Amplitude | 1.9 | 0.7 |
| Min Wavelength | 15.8 | 5.4 |
| Angle [deg] | 35.0 | 34.8 |

| IP2 Wave [mm] | As-Built | Scaled |
|---|---|---|
| 99% Amplitude | 9.0 | 4.5 |
| Mean Wavelength | 47.5 | 23.8 |
| Angle [deg] | 48.9 | 48.9 |

| OP4A Wave [mm] | As-Built | Scaled |
|---|---|---|
| Mean Amplitude | 1.9 | 0.7 |
| Mean Wavelength | 7.4 | 2.28 |
| Angle [deg] | 9.7 | 9.4 |

| IP3 Wave [mm] | As-Built | Scaled |
|---|---|---|
| Mean Amplitude | 3.7 | 1.9 |
| 10% Wavelength | 20.0 | 10.0 |
| Angle [deg] | 47.8 | 47.8 |

## 3 Coupon Manufacturing & Methodology

All test coupons consisted of four layer laminates infused utilizing a modified VARTM process with a PPG-Devold 1250 gram-per-square-meter primarily unidirectional E-glass and a Hexion RIM 135 resin system. The nominal fiber volume fraction of the panels was 55% with a nominal thickness of 0.8 mm for each layer resulting in a nominal total thickness of approximately 3.6 mm. Tensile coupons were cut to approximately 50 mm wide by 200 mm long and were tabbed resulting in a gage length of 100 mm. Compression coupons were cut to approximately 25 mm wide by 150 mm with gage lengths of 25 or 38 mm depending on flaw wavelength.

Manufacturing processes were developed and utilized to create coupons with wavy fibers (Riddle et al., 2013; Nelson et al., 2013). IP waves were introduced by manually pulling the fibers transversely for one entire wavelength. OP waves, also for one entire wavelength, were created by placing discontinuous fibers transversely to build up the waveform. Due to variability in the specimen manufacturing processes, it was necessary to characterize the as-built flaw parameters prior to testing to ensure that all correlations were performed accurately. Through thickness, IP wave images for each layer were collected with the use of a Computer Tomography (CT) scanner where wave parameters were measured as displayed in **Error! Reference source not found.**. Out-of-plane waves were measured with the use of a high fidelity digital photographs.

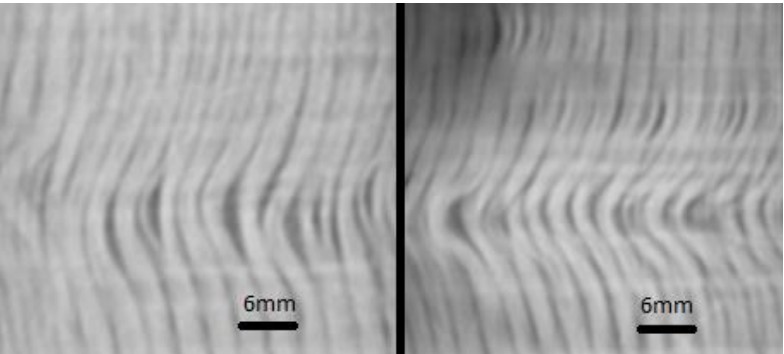

**Figure 8: Radiograph images of In-Plane waves found on different layers of one specimen.**





Scanning Electron Microscopy (SEM) was used to image the cut surface plane (**Error! Reference source not found.**). Image processing techniques were then used to identify the location and size of gas inclusions and ultimately calculate the planar area fraction of porosity. This value was then extrapolated to percent porosity by volume. Burn off testing was used to validate the percent porosity. However, this technique yields no indication of size or location of inclusion, therefore, it was not employed for data collection. Given the difficulty in testing and the destructive nature of this method, alternative methods continue to be investigated including radiodensity which has shown promising results (Shapurian, et al., 2006).

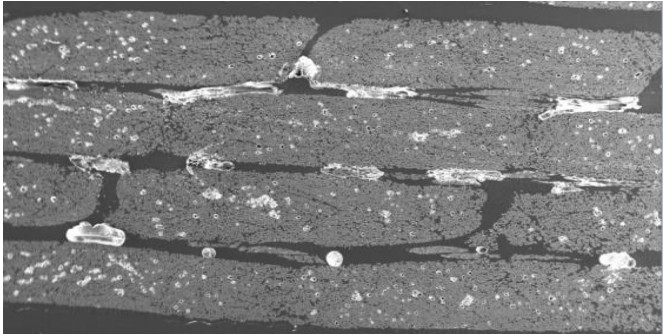

**Figure 9: Cross-section SEM image of a coupon containing porosity.**

Quasi-static, displacement controlled ramp tests on all specimens were conducted at a rate of 0.05 mm/s in tension and 0.45 mm/s in compression for all 4-ply coupons. These tensile tests were performed based on the ASTM D 3039 tensile testing of composites standard (ASTM D 3039, 2014). Compression testing was more loosely based on ASTM D 3410 and D 6641 (ASTM D 3410, 2014; ASTM D 6641, 2014). Digital Image Correlation (DIC) was utilized to capture displacement and full-field strain.

Material properties were calculated for each coupon and then averaged for each group. Where bending was found to be minimal enough to be disregarded, ultimate tensile or compressive strength was calculated:

$$F^{tu} = P^{max}/A \qquad (4)$$

where $F^{tu}$ is the ultimate tensile or compressive strength, $P^{max}$ is the maximum load before failure, and $A$ is the average cross sectional area. This equation was modified to calculate the stress at each point ($\sigma_i$), necessary for plotting of stress-strain curves, by substituting $P_i$, the load at the $i$th point, for $P^{max}$. Similarly, ultimate shear strength was calculated for ±45° specimen:

$$\tau_{12}^{tu} = P^{max}/2A \qquad (5)$$

where $\tau_{12}^{tu}$ is maximum in-plane shear.

Strain was calculated utilizing a DIC system based on the full field of the coupon such that it was calculated for the entire gage section. To ensure a consistent method that would allow for calculation for both unflawed controls and flawed specimens, strain was generalized for the entire gage length which was the same for all coupons. This allowed for consistent comparison given the different flaws, specifically the variation of fiber misalignment angles.



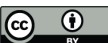

Once both stress and strain were calculated, modulus of elasticity (*E*) was calculated for each specimen utilizing this data. Initial linear portions of each stress-strain curve (generally 0.1-0.3% strain) were chosen to ensure accuracy and consistency of the chord modulus utilized:

$$E = \Delta\sigma/\Delta\epsilon \qquad (6).$$

## 4 Results & Discussion

### 4.1 Effects of Defects Trends

The goal of this work was to establish benchmark material and flaw testing based on in situ defect parameters to contribute to accurate prediction the Effects of Defects in thicker laminates such as those found in wind turbine blades. The results of failure stress verses average fiber angle for IP and OP waves are shown in **Error! Reference source not found.**. These results show that strength degradation in laminates with waves tend to correlate very well to the average off-axis fiber angle of all layers in the laminate as measured through thickness. An alternative correlation using the maximum fiber angle can be achieved with a minor reduction in accuracy. For example, an OP wave embedded in a planar structure under compression is predominately prone to buckling due to the inherent eccentricity. While buckling is a common mode of failure in a wind turbine blade, it is driven predominately by the global structure and local geometry effects. Thus, even with the use of symmetric OP waves to reduce buckling during coupon testing, the reduction in material property in a compressed section of a blade is likely to have a more complex effect.

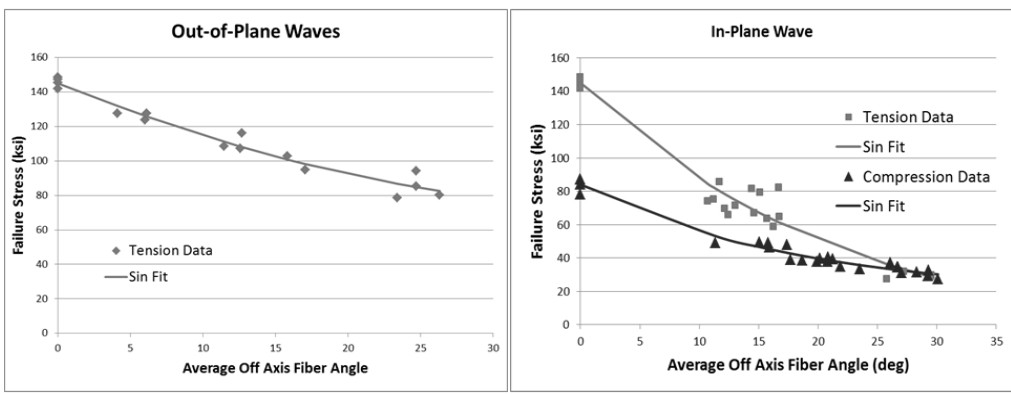

**Figure 10: Peak Stress of OP and IP Waves at various fiber angles (left-to-right, respectively).**

Linear regression analysis demonstrates that all the data displayed in **Error! Reference source not found.** fit best to exponentially decaying sinusoidal functions. This fit has roots from strength of materials failure criteria. For an off-axis ply, the stresses rotate according to:

$$\sigma'_{ij} = a_{ik} a_{jl} \sigma_{kl} \qquad (7)$$

where $\sigma'_{ij}$ is the rotated local stress and $\sigma_{kl}$ is the global stress, and $a_{ik}$, $a_{jl}$ are direction cosines of the rotated region. With an interactive failure criterion, such as Tsai-Wu, the failure curve verses off axis angle is essentially a decaying stress rotation function which starts with matrix dominated failure and quickly transitions to fiber dominated failure




with off-axis loading (Barbero, 2011). Based on these results, this type of analysis can be used to quickly assess the tension and compression failure strengths of wavy materials.

Due to variability in the specimen manufacturing processes, it was necessary to characterize the as-built flaw parameters prior to testing to ensure that all correlations are performed accurately. Through thickness, in-plane wave

5    images for each layer were collected with the use of a Computer Tomography (CT) scanner where wave parameters were measured as displayed in **Error! Reference source not found.**. For the case of IP waves, each layer's off axis fiber angle was recorded and examples of the layer-by-layer variation in fiber angle is given in **Error! Reference source not found.**. Once testing of the IP wave samples was completed, the results were reviewed for correlation to the maximum and average, through thickness wave angle. The analysis revealed very similar correlation traits. The

10   average wave angle proved to have better correlation and was used as the characteristic parameter to describe as-built flaw magnitudes. Out-of-plane waves were measured with the use of a high fidelity digital photographs and little migration during manufacturing was noted.

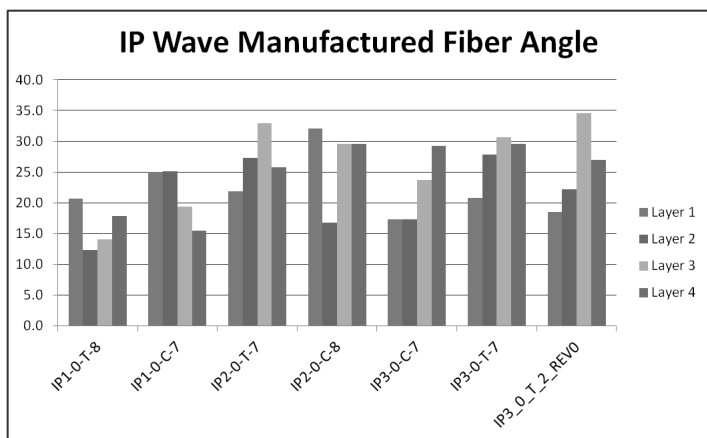

**Figure 11: Example of layer by layer characterization**

15   A vacuum bag technique was used to manufacture the test specimens in this investigation; therefore, it was necessary to include volume effects. To accomplish this, failure stresses were normalized by the part thickness (i.e. normalize fiber volume for the same amount of resin and fiber) allowing for direct assessment of strength as impacted by porosity. As shown in **Error! Reference source not found.**, porosity has a greater adverse effect on compressive strength than tensile strength due to the reduction of resin performance.




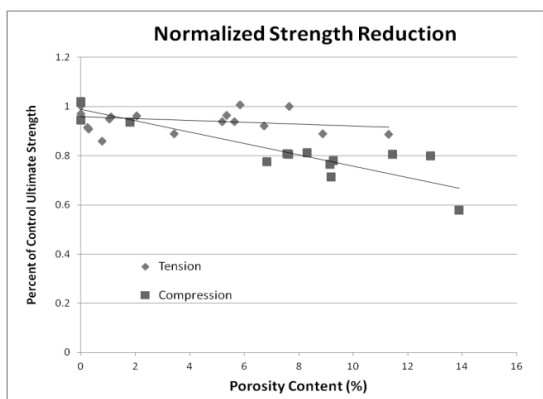

**Figure 12: Reduction in Strength due to porosity.**

Overall, the results of this testing effort determined correlation of ultimate strength to flaw characterization parameters exhibiting trends that were described through regression analysis with values for coefficients of determination greater than 0.95. Moreover, model predictions of load displacement curves were accurate to within ±5%.

## 4.2 Comparison of Material Properties

Material properties for each coupon were calculated and averaged for each flaw group and are shown in **Error! Reference source not found.** and **Error! Reference source not found.**. Standard deviations shown for ultimate stress indicate the consistency of each test. To ensure accuracy of these values, comparisons were made between the control results and the results of similar tests published in the Montana State University Composite Material Database (**Error! Reference source not found.**). Modulus of elasticity and maximum strain were chosen as points of comparison as they were critical for analytical inputs and correlation. These comparisons indicated that while the material properties compare acceptably for tension, the compressive failure strains for the test group appeared less accurate. This was likely due to the unrestrained method of testing in compression which resulted in bending and buckling. However, much less bending was noted in the flawed specimen, apart from OP waves. The lack of bending prior to failure indicated that damage was occurring at the flawed area prior to bending occurring suggesting that these data were acceptable.

**Table 3: Static properties for laminates tested in tension and calculated percentage of control laminates.**

| Tension | Control | | Porosity | | IP 1 | | IP 2 | | IP 3 | | OP 1 | | OP 2A | | OP 4A | |
|---|---|---|---|---|---|---|---|---|---|---|---|---|---|---|---|---|
| | 0° | ±45° | 0° | ±45° | 0° | ±45° | 0° | ±45° | 0° | ±45° | 0° | ±45° | 0° | ±45° | 0° | ±45° |
| Ultimate Stress (MPa) | 990 | 112 | 950 | 103 | 521 | 108 | 344 | 109 | 226 | 107 | 417 | 84 | 742 | 101 | 752 | 102 |
| *Standard Dev* | (40) | (2.0) | (19) | (1.5) | (24) | (4.1) | (43) | (1.1) | (24) | (4.7) | (26) | (5.3) | (79) | (2.2) | (43) | (2.8) |
| % Control | -- | -- | 96% | 93% | 53% | 97% | 35% | 98% | 23% | 96% | 42% | 75% | 75% | 91% | 76% | 91% |
| Strain at Failure (%) | 2.64% | 2.61% | 2.54% | 3.32% | 1.66% | 3.23% | 1.66% | 3.07% | 1.66% | 2.41% | 4.77% | 4.91% | 4.92% | 4.06% | 4.56% | 4.43% |
| % Control | -- | -- | 96% | 127% | 63% | 124% | 63% | 118% | 63% | 92% | 181% | 188% | 186% | 156% | 173% | 170% |
| Modulus of Elasticity (GPa) | 41.1 | 16.2 | 39.6 | 16.6 | 39.6 | 18.7 | 34.8 | 16.8 | 24.1 | 16.6 | 17.3 | 5.9 | 30.8 | 16.1 | 31.2 | 15.3 |
| % Control | -- | -- | 96% | 103% | 96% | 115% | 85% | 104% | 59% | 100% | 42% | 36% | 75% | 100% | 76% | 94% |
| Poisson's ratio | 0.27 | | -- | | -- | | -- | | -- | | -- | | -- | | -- | |

**Table 4: Static properties for laminates tested in compression and calculated percentage of control laminates.**



| Compression | Control | | Porosity | | IP 1 | | IP 2 | | IP 3 | | OP 1 | | OP 2A | | OP 4A | |
|---|---|---|---|---|---|---|---|---|---|---|---|---|---|---|---|---|
| | 0° | ±45° | 0° | ±45° | 0° | ±45° | 0° | ±45° | 0° | ±45° | 0° | ±45° | 0° | ±45° | 0° | ±45° |
| Ultimate Stress (MPa) | 582 | 124 | 491 | 125 | 257 | 165 | 216 | 181 | 216 | 139 | 95 | 43 | 227 | 90 | 207 | 86 |
| *Standard Dev* | (28) | (1.2) | (20) | (1.5) | (23) | (2.8) | (10) | (5.0) | (9.0) | (3.0) | (13) | (2.1) | (3.4) | (7.5) | (5.7) | (0.78) |
| % Control | -- | -- | 84% | 101% | 44% | 133% | 37% | 147% | 37% | 112% | 16% | 35% | 39% | 72% | 36% | 70% |
| Strain at Failure (%) | 1.76% | 1.16% | 1.44% | 1.06% | 0.84% | 0.59% | 0.84% | 0.51% | 0.92% | 0.82% | 0.70% | 1.11% | 1.04% | 0.94% | 0.92% | 0.84% |
| % Control | -- | -- | 82% | 91% | 48% | 51% | 48% | 44% | 52% | 71% | 40% | 96% | 59% | 81% | 52% | 72% |
| Est. Modulus of Elasticity (GPa) | 37.2 | 15.5 | 36.5 | 16.4 | 34.2 | 25.4 | 30.9 | 28.7 | 29.4 | 19.7 | 8.2 | 4.5 | 23.1 | 12.4 | 23.4 | 11.9 |
| % Control | -- | -- | 98% | 106% | 92% | 164% | 83% | 185% | 79% | 127% | 22% | 29% | 62% | 80% | 63% | 77% |
| Poisson's ratio | 0.28 | | -- | | -- | | -- | | -- | | -- | | -- | | -- | |

**Table 5: Comparison of control test results to published MSU composites database results in tension and compression. (\* indicates exact material match not available and a similar material system used.)**

| Test | Tension | | | | Compression | | | |
|---|---|---|---|---|---|---|---|---|
| Material Orientation | 0° | | ±45° | | 0° | | ±45° | |
| Data Source | CMD | BMT | CMD | BMT | CMD | BMT | CMD | BMT |
| Modulus of Elasticity (GPa) | 41.1 | 40.6 | 14.9 | 16.2 | 38.4* | 37.2 | 14.4* | 15.5 |
| Strain at Failure (%) | 2.7 | 2.6 | 2.9* | 2.6 | 2.4 | 1.8 | 1.6 | 1.6 |

### 4.2.1 IP Waves Strength and Stiffness

Ultimate stress values for the each of the 0° IP wave groups tested in tension were found to have a significant decrease in ultimate stress: 54% down to 25% of the control for waves IP1 through IP3, respectively. As noted in **Error! Reference source not found.**, the amplitude and wavelengths for each of these waves varied, and even though IP1 had the highest ultimate stress, it also had the largest amplitude. Furthermore, IP2 had a larger amplitude and wavelength than IP3, while the ultimate stress for each was approximately the same, though IP2 had a larger strain at failure than IP3. Based on previous research, similarity of the results was expected between the IP2 and IP3 groups, as the fiber angles were similar in the two groups.

It is also interesting to note that the stiffness for these groups was 85-96% of the control. Initial stiffness was similar to the control; however, the ultimate stresses and strains were notably lower. This was likely due to the load matrix "locking" the fibers into place at the ends of each wave before the matrix cracking noted above. Very similar results and trends were also noted for the 0° IP wave groups tested in compression. Overall, IP waves resulted in reduced material properties when included in 0° laminates.

The ±45° groups tested in tension were noted to have a similar damage progression as the 0° wave groups as noted in ultimate stress values very similar to the control group (96-98% of control) and the strains at failure were found to be relatively consistent with the control (92-112%). Of note was the stiffness increase compared to the control group (103-115%), likely for the same reasons given for the 0° IP groups noted above, which resulted in significantly lower values for Poisson's ratio. The ±45°compression results were rather remarkable, as the ultimate stress for all IP wave groups was significantly higher compared to similar control groups (127-185%) even though strains at failure were lower (48-52%). This resulted in significantly stiffer ±45° laminates causing a negative Poisson's ratio for the IP wave groups in compression. These results are due to the increase load-carrying ability of the laminates caused by the fibers in the wave approaching 0°. However, both IP2 and IP3 had the same fiber angle though the ultimate stresses in each were different: 181 and 139 MPa, respectively. This difference may be from unexpected responses during manufacture that resulted in differences between initial imparted amplitudes, wavelengths, fiber angles, and the fiber content of the final laminate as noted above. In short, while properties decreased in 0° laminates including





IP waves, laminates including ±45° performed as well or better than control, eliminating the need for further analysis. In addition, these data offer reasonable convergence points for the analytical models efforts.

### 4.2.2 Damage Progression

Use of the DIC system allowed for confirmation of the calculated strain and damage progression through strain field measurement during testing. Damage progression was found to vary for each defect type, but was observed to generally involve matrix cracking, ply delamination, load redistribution, and ultimately ply failure. Damage progression of the IP Waves, as shown in **Error! Reference source not found.**, was directly influenced by the flaw. With the aid of the DIC, it was noted that the strain accumulated in the wave area progressing transversely from the angled fiber toward the peak of the wave. Fiber breakage appeared to initiate at the point where the strain accumulations from each side of the wave met. These observations combined with the strains at failure indicate that damage accumulation was at lower strains than the control group and was the result of shear in the area of the wave. As noted and detailed below, the amount of shear was directly related to the magnitude of the fiber misalignment angle. These progressive damage data were intended as correlation points for analytical routines discussed.

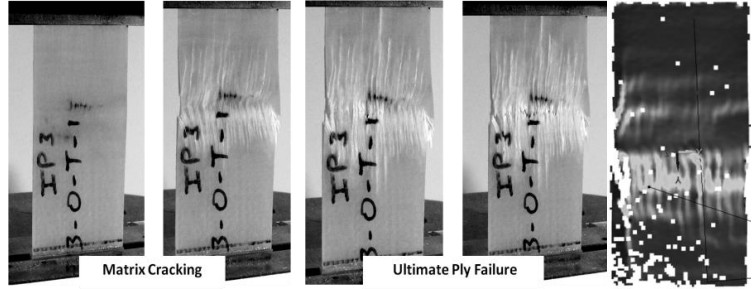

**Figure 13: Damage progression (left-to-right) and digital image correlation data (far-right) of IP Wave 1 with initial damage accumulating at the areas where fibers are not continuous along the length.**

In summary, it should be noted that the IP 1 case had decreases in material strength, and significant degradation was noted, making the result that this case was optimal for baseline use for modeling efforts (Riddle et al, 2017; Nelson et al, 2017). Further, it was decided that since this case had a fiber misalignment angle close to 30°, it would be a good median case for these endeavors. The resulting stress-strain curves, utilizing the DIC data from this test group, for this IP wave case in tension and compression are found in **Error! Reference source not found.**. Data beyond failure and maximum stress was gathered to begin to establish a comprehensive understanding of the material to be applied to future work with larger substructures and structures. As such, this geometry and these results were utilized as the baseline model for experimental/analytical correlation of each modeling type outlined below.





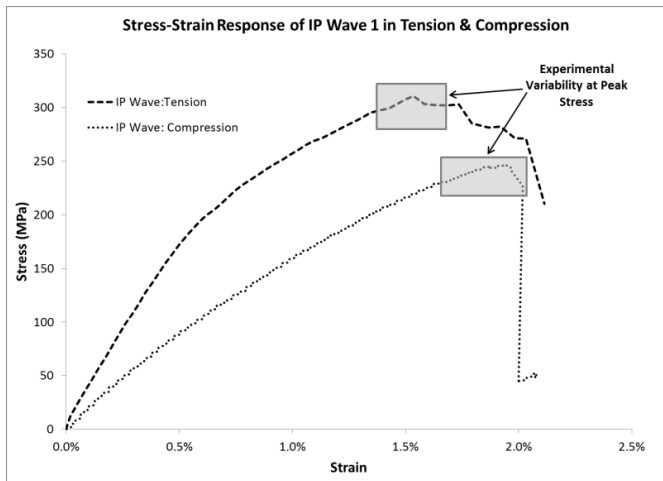

**Figure 14: Stress-strain of IP Wave 1 in tension and compression utilized for baseline model correlations with associated experimental variability.**

### 4.2.3 OP Waves

5    Test results for the OP wave groups are also noted in **Error! Reference source not found.** and **Error! Reference source not found.** with a representative stress-strain response shown in **Error! Reference source not found.**. Results from test observation and the DIC suggest that each of the OP wave groups was noted to have similar damage progression. Also, like the behavior of IP waves, as strain levels increased, cracks initiated in the resin between the layers at the ends of the wave before delaminating. However, unlike the behavior of IP waves, after delamination and

10    significant fiber straightening, the failure area for the OP wave specimens was concentrated at the peak area of the wave. This was due to the fibers being pulled straight and the center of bending being at the peak of the wave. It must be noted that the wave forms for all the OP 1 group delaminated during testing. This resulted in an extreme decrease in the ultimate stress and stiffness results of the OP1 groups in both tension and compression. As such, OP 1 was deemed unusable for correlation in both tension and compression.




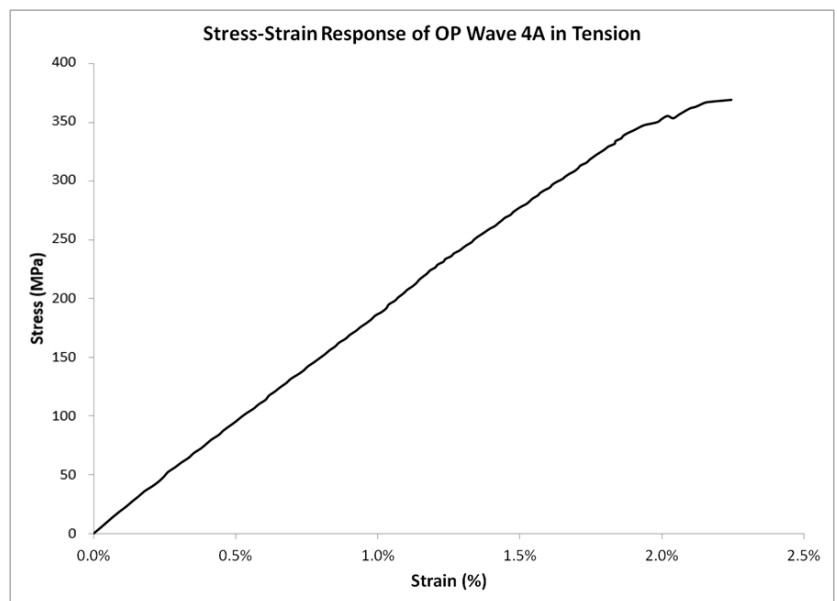

**Figure 15: Stress-strain of OP Wave 4A in tension utilized for initial OP wave model correlations.**

The two other cases, OP 2A and OP 4A, were found to have a more consistent response. Ultimate stress and strain at failure values for the OP 2A and OP 4A 0° and ±45° tension groups were decreased compared to the control but were

increased compared to the IP waves. Thus, moduli of elasticity values were similar to the control due to load being transferred more consistently through the wave than seen with IP waves due to the configuration described above. Given the consistency of these waves in tension, the OP 4A case was utilized for correlation. However, compression testing of the OP waves proved to be very difficult, as large wavelengths necessitated a long unsupported gage length. This resulted in significant bending as the load transferred through the wave. As such, significant decreases in

calculated moduli of elasticity, ultimate strength, and strain at failure were noted and results were considered unusable for correlation given these responses. Overall, the static testing performed allowed for initial analysis while determining convergence points for analytical models.

## 5 Conclusions and Future Work

Using this consistent framework that was established and validated, defects common to wind turbine blades have been

quantified. To effectively characterize, categorize, and analyze defects, the frame requires accurate data collection following consistent scientific procedures. With proper characterization, it is possible to establish the mechanical response of a flaw using laboratory testing. Results from static testing indicate that there is a strong correlation between flaw parameters and mechanical response. Since the flaws went across the entire width of the sample, applying these knockdowns directly is conservative, but may not be realistic, especially if surrounding material in a blade structure

can redistribute loads from local failures. Going forward, the characterization techniques described herein may be applied to incoming data will enable the generation of a statistically significant and comprehensive flaw database.



This work provides a sound starting point, but only constitutes the building blocks for a comprehensive reliability program aimed at reducing failures as a result of defects. Since reliability estimation is inadequate for composite structures due to the uncertainties, a probabilistic approach is required to achieve an acceptable level of confidence. This approach must consider multi-scale mechanical property variability, damage/defect detection, damage

5   progression, residual strength analysis, global, and macro structural response

Using the metrics developed herein to precisely address the geometric nature of flaws based on statistical commonality in blades, mechanical testing and probabilistic modeling were performed. The work herein led to establishment a consistent framework that was validated for quantitative categorization and analysis of flaws to predict blade failure. Further, this significant coupon level testing effort has determined material properties and characterized damage

10  progression in both flawed and unflawed specimen allowing for baseline comparisons of the modeling methods. In short, these data allowed for direct comparison in determination of the consistency, accuracy, and predictive capability of each modeling approach.





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
