# Peer review of "Characterization and Mechanical Testing of Manufacturing Defects Common to Composite Wind Turbine Blades"

_Wind Energy Science, 2017_

## Referee Comment (RC1) · Anonymous Referee #1 · 28 May 2017

The manuscript addresses defects in composites in the form of fibre waviness. The characteristic fibre waviness observed in wind turbine blades is down-scaled to coupon test specimens, and the effect on mechanical properties is measured and analysed. The topic is relevant, and the experimental approach of the study is interesting and useful. There is however a number of major and minor critical things that need to be addressed before the manuscript can be recommended for publication.

- Figures 1 and 2 are central for the understanding of the approach taken with the measurements of OP and IP waves. These figures must be improved, e.g. by including a schematic drawing of a materials specimen in 3D showing the two types of waves. Also, the ruler representation must be identical. The edge of the material in Figure 2

should be seen, to be able to identify the difference between OP and IP.

- At page 6, the equation (3) is correct and relevant. It is however not clear how and where the equation is used to scale the fracture strengths of the composites. This must be made clear.

- Table 2. It is not clear how the fibre angles are scaled down to coupon test specimens. There are several errors in the table, e.g. 1.9 mm instead of 1.7 mm, and 7.4 mm instead of 67.4 mm. It is also confusing that OP4A and IP1 are the two baseline settings of mean values, and then placed in the bottom and top of the table, respectively.

- It is not clear how the induced off-axis fibre angles in the specimens correspond to the wanted ones given in Table 2. This needs to be made clear.

- In general, throughout the manuscript, there are error messages at places where references are made to figures and tables.

- At page 4, line 3, it is not clear how the off-axis angle is determined. This must be explained.

- Figures 5 and 6. In each figure, the axes of the two diagrams should be the same, to be able to directly compare the magnitude of the OP and IP waves. It could be considered to show the results of OP and IP waves in the same diagram.

- Figure 6. Why are the results sorted from low to high by observations?

- It can be argued that the use of 3 figures and 1 table for presenting the results of the measured OP and IP waves is too much. It is recommended omitting Figure 6.

- Page 7, line 16, it is not clear how digital photographs were used to measure OP waves. Based on polished cross-sections?

- Figure 8, "Radiographic images"?

- Figure 9, the porosity must be indicated, are they shown by black or white spots?

- Page 9, line 9-11. It is said that the strength degradation correlates well with the average off-axis fibre angle. It would be more correct to say that the correlation between the two parameters is well described by a sine function. Later it is said that the average fibre angle is showing a better correlation than the maximum angle; how to see that?

- Figure 10. Use SI units. The fitting lines in (b) seem to be consisting of two lines (with transition points at about 10 and 13 degrees), and not made with a continuous mathematical function.

- Figure 10. There are no compression data in (a)?

- At page 9, last line, change to "start with fibre dominated failure and transit to matrix dominated failure mode"?

- Figure 11 shows a rather large variation in fibre angles between the 4 layers. This variation must somehow be included in the analysis as an uncertainty parameter.

- At page 10, line 16, it is said that failure stress is normalised by the part thickness. It is not clear how this is done, it needs to be explained.

- Figure 12, It seems that both lines should start in 1. There are many samples with really high content of porosity, up to 14%. How is it ensured that the effect of porosity is not larger than the effect of the OP and IP waves?

- Page 11, line 5, it is said that model predictions of load displacement curves were accurate to within +/-5%; where is that shown?

- Page 11, line 9, "Standard deviations shown for ultimate stress indicate the consistency of each test", what is meant by that?

- Table 3. What is meant by the "Porosity" column?

- Table 5, what are "CMD" and "BMT"?

- In the introduction, p. 1, line 27-28, the terms "advanced composite materials" and

"lower cost composite materials" seem to be in contradiction to each other.

- In affiliations, country is missing.

---

## Referee Comment (RC2) · Anonymous Referee #2 · 21 Jun 2017

Overall: The paper deals with relevant scientific and industrial topics: mechanical performance of wrinkles and porosity in composite laminates used for wind turbine blade application. The structure of the paper is overall good and well written. Experimental testing is well conducted.

Specific remarks: - All figure references are not showing correctly. - Fig 1: What is the natural waviness of a ply compared to the measured fibre misalignment? - Fig 4. Are you using the maximum misalignment angle for comparison? If so, then clarify remaining plots to reflect maximum angle and not just 'Fibre angle'. If not, then elaborate what fibre angle is used. - P2 Line 12: where is the field data originating from? Blade cut

outs? Can these data be public assessed? - Fig 7: It is unclear whether you choose to using normal or Weibull distribution going forward. - Eq. 3: what is the value of Weibull modulus, m? It is a good idea to use the scale effect to design test samples. - P9 Line 6: Did you succeed in finding your goal? Please elaborate. - Fig 10 cmt 1: It is a shame the compression results for OP waves are not presented as these are probably the most interesting. Can it be included in spite of the variation? - Fig 10 cmt 2: It is unclear how the sine-fit is generated, please elaborate. The fit originates from Eq (7), but the connection to Fig. 10 is not obvious. - Fig 10 cmt 3: Can the plot be shown as a relative strength knockdown, ie. normalised by pristine strength? - P10 Line 18: Elaborate on why porosity has larger effect on compression than tension? - Fig 12: What is a typical level of porosity using a VARTM process? I imagine that >10% would indicate unwetted fibres, and can no longer be considered as porosity. Looks like the fit to compression data is quite vague, what is the correlation coefficient for the fits? Is there even a valid trend? What are the findings from other studies testing porosity? - Section 4.2.2: The content of this section is relative thin. Can this section be elaborated and show some DIC strain plots of the various strain components for instance? - Fig 13: Relate the damage stages to the stress-strain curve or at least state the load fraction at the given spot. The DIC plot to the far right is useless; no scalebar is shown and it does not state which strain component is shown. - P14 Line 9ff: For OP waves it is unclear what the failure mode is; is it interlaminar delamination or buckling? What is the failure mode in compression? - References: align reference list as per journal standard (italic font, use of capital letters, etc).

General remarks: - Have you considered fatigue? - What is the effect of wrinkles on blade reliability? - How are 'common defects' handled in the blade strength design? - The method used for identifying the wrinkles is destructive, any ideas on a non-destructive evaluation method?

---

## Author Response (AR1)

**Author Comments for Review #1**

The authors are grateful for the comments, suggestions, and insight from the reviewer. Please find comments below [with text locations included where appropriate] and an updated version of the paper attached with all changes marked and notated.

**RC1**  Figures 1 and 2 are central for the understanding of the approach taken with the measurements of OP and IP waves. These figures must be improved, e.g. by including a schematic drawing of a materials specimen in 3D showing the two types of waves. Also, the ruler representation must be identical. The edge of the material in Figure 2 should be seen, to be able to identify the difference between OP and IP.

    *AR*  Figures 1 & 2 have been replaced with a new schematic drawing, identifying the wave types and their orientation relative to the entire blade structure. The wave image in Figure 1 has been added to Figure 3 to help clarify the discretization process and helps readers understand their relationship to wind turbine blade structures.

**RC2**  At page 6, the equation (3) is correct and relevant. It is however not clear how and where the equation is used to scale the fracture strengths of the composites. This must be made clear.

    *AR*  Significant detail added to the text describing the use of Equation used. [p. 6]

**RC3**  Table 2. It is not clear how the fibre angles are scaled down to coupon test specimens. There are several errors in the table, e.g. 1.9 mm instead of 1.7 mm, and 7.4 mm instead of 67.4 mm. It is also confusing that OP4A and IP1 are the two baseline settings of mean values, and then placed in the bottom and top of the table, respectively.

    *AR*  Table 2 reformatted for clarity and numbering scheme updated throughout as suggested.

**RC4**  It is not clear how the induced off-axis fibre angles in the specimens correspond to the wanted ones given in Table 2. This needs to be made clear.

    *AR*  Original blade data removed as it was redundant to Table 1 and Figure 7 above.

**RC5**  In general, throughout the manuscript, there are error messages at places where references are made to figures and tables.

    *AR*  All links fixed.

**RC6**  At page 4, line 3, it is not clear how the off-axis angle is determined. This must be explained.

    *AR*  Added "…which is found by measuring the maximum misalignment angle of deviation from the intended fiber direction" to clarify. [p. 4, lines 12-14]. This provides the reader with an unambiguous description of the procedure.

**RC7**  Figures 5 and 6. In each figure, the axes of the two diagrams should be the same, to be able to directly compare the magnitude of the OP and IP waves. It could be considered to show the results of OP and IP waves in the same diagram.

    *AR*  Figure 6 has been removed and Figure 5 has been consolidated as suggested.

**RC8**  Figure 6. Why are the results sorted from low to high by observations?

    *AR*  Figure 6 has been removed as it was deemed redundant.

**RC9**  It can be argued that the use of 3 figures and 1 table for presenting the results of the measured OP and IP waves is too much. It is recommended omitting Figure 6.

    *AR*  Figure 6 has been removed as it was deemed redundant.

**RC10**  Page 7, line 16, it is not clear how digital photographs were used to measure OP waves. Based on polished cross-sections?

    *AR*  OP specimens were not polished; porosity specimen were. Detail added to clarify. [p. 7, line 20-21 and p.8, line 1-2]

**RC11**  Figure 8, "Radiographic images"?

***AR*** Updated.  Thanks!

***RC12*** Figure 9, the porosity must be indicated, are they shown by black or white spots?

    ***AR*** Added clarification to caption to identify the porosity as the white spots.

***RC13*** Page 9, line 9-11. It is said that the strength degradation correlates well with the average off-axis fibre angle. It would be more correct to say that the correlation between the two parameters is well described by a sine function. Later it is said that the average fibre angle is showing a better correlation than the maximum angle; how to see that?

    ***AR*** Sentences have been reworded to clarify: "These results show that strength degradation in laminates with waves tend to correlate well with the average of the maximum fiber misalignment angles of all layers in the laminate as measured through the thickness. An alternative correlation using the single maximum fiber angle can be achieved with a minor reduction in accuracy." [p. 9, line 11-14]

***RC14*** Figure 10. Use SI units. The fitting lines in (b) seem to be consisting of two lines (with transition points at about 10 and 13 degrees), and not made with a continuous mathematical function.

    ***AR*** Thanks!  Also, additional data points added to ensure consistency of trendline

***RC15*** Figure 10. There are no compression data in (a)?

    ***AR*** Given the inability to scale compression of the OP Wave in a representative manner, we are not comfortable presenting the results. We believe the data would not be relevant. This has been more directly explained in the text. [p. 9 line 14-19]

***RC16*** At page 9, last line, change to "start with fibre dominated failure and transit to matrix dominated failure mode"?

    ***AR*** Nice catch; thanks!

***RC17*** Figure 11 shows a rather large variation in fibre angles between the 4 layers. This variation must somehow be included in the analysis as an uncertainty parameter.

    ***AR*** Given the nature of the work and the focus of the BRC, it was determined that the worst-case was more important for characterization.  In addition, due to the number of uncertainty parameters considered in the companion paper (Riddle et al.), this worst-case method was deemed sufficient. The authors will keep this comment in mind when considering future work.

***RC18*** At page 10, line 16, it is said that failure stress is normalised by the part thickness. It is not clear how this is done, it needs to be explained.

    ***AR*** Details were confused in the reduction of the work into this paper.  Clarification has been made to this entire section. [Section 4.1.2: p. 10 line 22 through p. 11 line 9]

***RC19*** Figure 12, It seems that both lines should start in 1. There are many samples with really high content of porosity, up to 14%. How is it ensured that the effect of porosity is not larger than the effect of the OP and IP waves?

    ***AR*** Trends fixed to start at 1.  The porosity section has been updated and refocused to more directly represent the results.  Combined effects of defects were beyond the scope of this study and has been noted as future work.

***RC20*** Page 11, line 5, it is said that model predictions of load displacement curves were accurate to within +/-5%; where is that shown?

    ***AR*** This was a misplaced sentence from previous editing and did not belong in this section. It has been removed and may be found in the companion paper.

***RC21*** Page 11, line 9, "Standard deviations shown for ultimate stress indicate the consistency of each test", what is meant by that?

*AR* Clarification text added: "…standard deviations included for ultimate stress to indicate the distribution size of the coupons tested for each defect type." [p. 11 line 11-13]

*RC22* Table 3. What is meant by the "Porosity" column?

*AR* Column header changed from "Porosity" to 2% Porosity" in both Tables 3 & 4.

*RC23* Table 5, what are "CMD" and "BMT"?

*AR* Table 5 updated with Data Source names changed for clarity.

*RC24* In the introduction, p. 1, line 27-28, the terms "advanced composite materials" and "lower cost composite materials" seem to be in contradiction to each other.

*AR* Clarified by changing to "continuous fiber" and "Lower cost fiberglass materials" respectively [p. 1, line 27-30]

*RC25* In affiliations, country is missing.

*AR* Affiliations added

**Author Comments for Review #2**

The authors are grateful for the comments, suggestions, and insight from the reviewer. Please find comments below [with text locations included where appropriate] and an updated version of the paper attached with all changes marked and notated.

*Specific remarks:*

**RC1** All figure references are not showing correctly.

> **AR** It is unclear why the original upload had these discrepancies. However, it has been fixed throughout. Thanks!

**RC2** Fig 1: What is the natural waviness of a ply compared to the measured fibre misalignment?

> **AR** The natural waviness has been considered to be zero throughout this work due to use of uni-directional materials as prescribed by the BRC. Clarity has been added to identify the use of uni-directional material. [p. 2, line 20-23]

**RC3** Fig 4. Are you using the maximum misalignment angle for comparison? If so, then clarify remaining plots to reflect maximum angle and not just 'Fibre angle'. If not, then elaborate what fibre angle is used.

> **AR** Clarifying text has been added throughout the paper to ensure clarity on the exact meaning of "fiber angle" at any given point.

**RC4** P2 Line 12: where is the field data originating from? Blade cut outs? Can these data be public assessed?

> **AR** More description added here and particularly in the first paragraph of Section 2 [p 2, line 20-22]. Also, a reference to SANDIA report 1 has been added for public access

**RC5** Fig 7: It is unclear whether you choose to using normal or Weibull distribution going forward.

> **AR** Added text [p. 5, line 12-14] In general, these distributions were presented for work with probabilistic effects of defects. It is up to the reader to decide which is best for his/her case.

**RC6** Eq. 3: what is the value of Weibull modulus, m? It is a good idea to use the scale effect to design test samples.

> **AR** Added with justification of value along with clearer explanation of use [p. 6, line 9-21]

**RC7** P9 Line 6: Did you succeed in finding your goal? Please elaborate.

> **AR** Reworded and added additional text for clarity. [p. 9, line 6-9]

**RC8** Fig 10 cmt 1: It is a shame the compression results for OP waves are not presented as these are probably the most interesting. Can it be included in spite of the variation?

> **AR** Given the inability to scale compression of the OP Wave in a representative manner, we are not comfortable presenting the results. In particular, there is too much dependence on geometry for the results. Consequently, we believe the data would not be relevant. This has been more directly explained in the text. [p. 9 line 14-19]

**RC9** Fig 10 cmt 2: It is unclear how the sine-fit is generated, please elaborate. The fit originates from Eq (7), but the connection to Fig. 10 is not obvious.

> **AR** Further clarified. [p. 11, line 1-8]

**RC10** Fig 10 cmt 3: Can the plot be shown as a relative strength knockdown, ie. normalised by pristine strength?

> **AR** Great suggestion. Thanks!

**RC11** P10 Line 18: Elaborate on why porosity has larger effect on compression than tension?

> **AR** The entire porosity section has been reworked to better tell the story of the results pertinent to the rest of the work contained herein. As such, a narrower window of data has been assessed whereby the tension and compression trends are similar. [p. 10 line 22 through p. 11 line 9]

*RC12* Fig 12: What is a typical level of porosity using a VARTM process? I imagine that >10% would indicate unwetted fibres, and can no longer be considered as porosity. Looks like the fit to compression data is quite vague, what is the correlation coefficient for the fits? Is there even a valid trend? What are the findings from other studies testing porosity?

> *AR* Porosity figures have been updated as well [Figure 10] as noted in previous comment. Specifically, we identify that other than just % porosity, there are distribution and size considerations that are not considered herein, but could be part of future work.

*RC13* Section 4.2.2: The content of this section is relative thin. Can this section be elaborated and show some DIC strain plots of the various strain components for instance?

> *AR* Great suggestions. Thanks! [p. 13 line 3-23]. This was added and shows the progression to add understanding for the reader.

*RC14* Fig 13: Relate the damage stages to the stress-strain curve or at least state the load fraction at the given spot. The DIC plot to the far right is useless; no scalebar is shown and it does not state which strain component is shown.

> *AR* Again, great suggestions. [Figure 11] Thanks!

*RC15* P14 Line 9ff: For OP waves it is unclear what the failure mode is; is it interlaminar delamination or buckling? What is the failure mode in compression?

> *AR* Additional clarity added [p. 14, line 5-15]. In particular, the progressive damage and final failure are described

*RC16* References: align reference list as per journal standard (italic font, use of capital letters, etc).

> *AR* Completed.

**General remarks:**

*AR* [Each of these remarks have been addressed briefly in the manuscript as well. Good, insight remarks. Thanks!]

*1)* Have you considered fatigue?

> a. Yes, it is discussed in the companion (wes-2017-14) as it pertains to the results from this work being translated into the blade study. A separate paper covering this is anticipated.

*2)* What is the effect of wrinkles on blade reliability?

> a. Good question. These data are helpful and the same connection to the companion papers through the blade test offer some insight. As a historical perspective, the BRC decided to not address wrinkles (defined as a fold-back reversal of a wave) because wrinkles are an uncceptable manufacturing defect. Stay tuned; more should be coming.

*3)* How are 'common defects' handled in the blade strength design?

> a. Typically, through the use of knockdown factors, potentially adding material, weight, and time. This scalar knockdown is probably conservative in most areas, but could be non-conservative in regions of high risk. Treating manufacturing defects in a quantitative manner such as in this work provides more insight that scalar safety factors applied over the entire blade. The present work also provides a rational basis for replacing safety factors with probabilistic design and certification approaches.

*4)* The method used for identifying the wrinkles is destructive, any ideas on a nondestructive evaluation method?

> a. Several methods of NDE to investigate this have been taken on by other member of the BRC as they have more experience in this area. In the BRC, there is a Probability of Detection (POD) activity for finding and quantifying manufacturing defects. This work is a study of available and practical inspection methods wind turbine blades.

Marked Up Manuscript Version

[revised manuscript text omitted]

**Commented [u7]:** What is natural fiber waviness? (REV #2 COMMENT -2)

**Commented [n8R7]:** Added clarity in text. The natural waviness is negligible due to use of uni-directional materials per the BRC

Figure 1: Example of an out-of-plane (OP) wave where fiber waviness is observed through the thickness.

[Figure]

**Commented [n9]:** Figures 1 and 2 are central for the understanding of the approach taken with the measurements of OP and IP waves. These figures must be improved, e.g. by including a schematic drawing of a materials specimen in 3D showing the two types of waves. Also, the ruler representation must be identical. The edge of the material in Figure 2 should be seen, to be able to identify the difference between OP and IP.
(REV #1 COMMENT -1)

**Commented [n10R9]:** Added new image based on this comment

**Commented [u11]:** Update all figure references throughout (REV #2 COMMENT -1)

**Commented [n12R11]:** Addressed in final version

**Commented [n13]:** In general, throughout the manuscript, there are error messages at places where references are made to figures and tables.
(REV #1 COMMENT -5)

**Commented [n14R13]:** Addressed in final version

[revised manuscript text omitted]

**Commented [n15]:** At page 4, line 3, it is not clear how the off-axis angle is determined. This must be explained.
(REV #1 COMMENT -6)

**Commented [n16R15]:** Added "which is found by measuring the maximum misalignment angle of deviation from the intended fiber direction"

**Commented [u17]:** Are you using the maximum misalignment angle for comparison? If so, then clarify remaining plots to reflect maximum angle and not just 'Fibre angle'. If not, then elaborate what fibre angle is used.
(REV #2 COMMENT -3)

**Commented [u17]:** Are you using the maximum misalignment angle for comparison? If so, then clarify remaining plots to reflect maximum angle and not just 'Fibre angle'. If not, then elaborate what fibre angle is used.
(REV #2 COMMENT -3)

**Commented [n18R17]:** To clarify throughout the term "Maximum fiber misalignment angle" has been used with Figures 7,10,11 and Tables 1,2 being updated (old numbers)

**Commented [u19]:** where is the field data originating from? Blade cut outs? Can these data be public assessed?
(REV #2 COMMENT -4)

**Commented [n20R19]:** More description added here and particularly in the first paragraph of Section 2 (p 2, line 20-22)

Also a reference to SANDIA report 1 has been added for re: public access

[Figure]

**Figure 4: Collected OP and IP wave data (left-to-right, respectively).**

[Figure]

 Error! Reference source not found. **(left-right, respectively).**

Mean and standard deviation values were used to develop Normal distributions to describe the frequency of flaw magnitude occurrences and Weibull distributions (2 parameter) were generated using Maximum Likelihood Estimation. To develop frequency of occurrences distributions, the off-axis fiber angle values gathered from all wave segments were binned together into groups as shown in **Error! Reference source not found.**. For OP waves, angles were binned in one-degree increments while for IP waves, angles were binned in four-degree increments. The frequency of each fiber angle can be seen in **Error! Reference source not found.** where the observed frequency of occurrence is displayed with applicable Weibull and Normal distribution curves. In general, wave fiber angle values show a strong

**Commented [n21]:** Figures 5 and 6. In each figure, the axes of the two diagrams should be the same, to be able to directly compare the magnitude of the OP and IP waves. It could be considered to show the results of OP and IP waves in the same diagram (REV #1 COMMENT -7)

**Commented [n22R21]:** Great suggestion. Figure 5 updated accordingly

**Commented [n23]:** Figure 6. Why are the results sorted from low to high by observations?
(REV #1 COMMENT -8)

It can be argued that the use of 3 figures and 1 table for presenting the results of the measured OP and IP waves is too much. It is recommended omitting Figure 6.
(REV #1 COMMENT -9)

**Commented [n24R23]:** The authors agree with the comment #9 and Figure 6 has been removed.

inclination towards common distributions such as the Weibull and Normal distributions for both cases with the Normal distribution utilized throughout the probabilistic analysis.  binning procedures where applied to amplitude and wavelength data for both wave types; however, the distributions were less accurate further justifying characterization with fiber angle. Generalized information regarding both IP and OP wave group data is summarized in **Error! Reference source not found.**.

[Figure]

**Figure 5: Distribution of all OP and IP wave fiber angles gathered  (left-to-right, respectively).**

**Table 1: Summary of wave data generated from processes outline above.**

| OP Waves | Amplitude (cm) | Wavelength (cm) | Maximum Fiber Angle (deg) |
|---|---|---|---|
| **Min** | 0.02 | 1.58 | 0.6 |
| **Max** | 0.85 | 21.49 | 39.0 |
| **Mean** | 0.17 | 6.74 | 8.6 |
| **Standard Deviation** | 0.11 | 3.00 | 2.8 |
| **IP Waves** | Amplitude (cm) | Wavelength (cm) | Maximum Fiber Angle (deg) |
| **Min** | 0.11 | 1.08 | 8.7 |
| **Max** | 1.18 | 28.12 | 50.7 |
| **Mean** | 0.37 | 4.75 | 26.7 |
| **Standard Deviation** | 0.23 | 4.96 | 9.3 |

A test program focused on the characterization of the mechanical performance of specimen with scaled flaws was developed from the data collected in this study. Given the scale difference between the blades and coupon-sized test specimen, the flaws were scaled using a Weibull scaling analysis where fracture strength was adjusted with material volume. Based on this "weakest link" theory, as material volume decreases, the population of defects also decreases, thereby reducing the probability of a failure from a flaw. The ratio of fracture strengths may then be found if the probability of survival is assumed to be the same for both small and large-scale composite structures:

$$\frac{\sigma_1}{\sigma_2} = \left(\frac{V_2}{V_1}\right)^{\frac{1}{m}} \tag{3}$$

where $\sigma_{1,2}$ are the fracture strengths, $V_{1,2}$ are the volumes and $m$ is the Weibull modulus. Comparisons were made between the coupons and as-built blade sections, utilizing the same length (coupon gauge length) and unit width. The 4-ply laminate test specimen have a thickness of ~3.2mm which is 8.8 times smaller (volumetrically) than actual as-built sections. Using the volume fraction and a modulus of 29.1 in Equation 3 (Wisnom, 1999), the Weibull scaling expression, it was found that the fracture strength for the larger as-built blade sections was expected to be
* * *
**Commented [u25]:** It is unclear whether you choose to using normal or Weibull distribution going forward. (REV #2 COMMENT -5)

**Commented [n26R25]:** Normal used in Part B (wes-2017-14) therefore discussion of it is found there

**Commented [u27]:** what is the value of Weibull modulus, m? It is a good idea to use the scale effect to design test samples. (REV #2 COMMENT -6)

**Commented [n28R27]:** Addressed to include value and justification for its use.

**Commented [n29]:** At page 6, the equation (3) is correct and relevant. It is however not clear how and where the equation is used to scale the fracture strengths of the composites. This must be made clear. (REV #1 COMMENT -2)

**Commented [n30R29]:** Additional detail added in following paragraph.

approximately 7.1% less than the coupons.   To scale the as-built OP flaw waveforms, the mathematical description of each wave was integrated over the half wavelength to calculate the cross-sectional area bounded by each flaw curve. This was the only parameter needed as unit width was considered. The volumetric ratio between the full-scale blade sections and test specimen was then applied to the as-built flaw cross sectional area. Knowing the scaled cross sectional area, the amplitude and wavelength of each wave was solved for. It is important to note that this analysis was appropriate for the out-of-plane waves only. The in-plane waves did not vary with thickness, and therefore, a volumetric scaling approach was not taken. Instead, each was scaled by the same ratio to fit within the coupon dimensions.

Using this method of scaling, three wave forms for each type of flaw were systematically chosen, as shown in **Error! Reference source not found.**, for testing based on geometry characterization and statistical significance while representing data points around an angular region of interest. The parameters for waves OP1 and OP2 were identified to be included due to the similarity in fiber angle occurring from different a different combination of amplitude and wavelength. The additional OP wave (OP3) had mean values for all three parameters, and therefore, landed in the center of all the parameter distributions.  -As such, these data points combined to sufficiently described an OP wave common to the specific wind turbine application. The reader may note subtle variations in the mean values when compared to Table 1 which result from the scaling process. By design, the mean value also delivered baseline values for comparison of the effects of amplitude and wavelength independently with the OP1 and OP2 results. In-plane test waves IP1 and IP2 followed this same approach as the OP where they each had different amplitudes and wavelengths, but resulted in similar misalignment angles. Similarly, the IP3 case represented the parametric mean for all values.

Table 2:  OP and IP wave parameters as scaled for  in the coupon testing program.

| Scaled OP Waves | Amplitude (cm) | Wavelength (cm) | Maximum Fiber Angle (deg) |
|---|---|---|---|
| OP1 | 0.29 | 2.28 | 36.8 |
| OP2 | 0.07 | 0.54 | 34.8 |
| OP3 | 0.07 | 0.23 | 8.6 |
| **Scaled IP Waves** | **Amplitude (cm)** | **Wavelength (cm)** | **Maximum Fiber Angle (deg)** |
| IP1 | 0.45 | 2.38 | 48.9 |
| IP2 | 0.19 | 1.00 | 47.8 |
| IP3 | 0.19 | 2.38 | 24.8 |

**Commented [n31]:** Table 2. It is not clear how the fibre angles are scaled down to coupon test specimens. There are several errors in the table, e.g. 1.9 mm instead of 1.7 mm, and 7.4 mm instead of 67.4 mm. It is also confusing that OP3 and IP3 are the two baseline settings of mean values, and then placed in the bottom and top of the table, respectively.
(REV #1 COMMENT -3)

It is not clear how the induced off-axis fibre angles in the specimens correspond to the wanted ones given in Table 2. This needs to be made clear.
(REV #1 COMMENT -4)

**Commented [n32R31]:** REV #1 COMMENT -3: Table reformatted and number scheme reconsidered throughout.

-4: Original blade data removed as it was redundant to Table 1 and Figure 7 above.

**3 Coupon Manufacturing & Methodology**

[revised manuscript text omitted]

| **Commented [u33]:** Did you succeed in finding your goal? Please elaborate. (REV #2 COMMENT -7) |
| **Commented [n34R33]:** Reworded sentence and added another. |

**4.1.1 IP and OP Wave Trends**

The results of failure stress verses average fiber angle for IP and OP waves are shown in Figure 8. These results show that strength degradation in laminates with waves tend to correlate well with the average  fiber misalignment angles of all layers in the laminate as measured through the thickness. An alternative correlation using the single maximum fiber angle can be achieved with a minor reduction in accuracy. For example, an OP wave embedded in a planar structure under compression is predominately prone to

| **Commented [n35]:** It is said that the strength degradation correlates well with the average off-axis fibre angle. It would be more correct to say that the correlation between the two parameters is well described by a sine function. Later it is said that the average fibre angle is showing a better correlation than the maximum angle; how to see that? (REV #1 COMMENT -13) |
| **Commented [n36R35]:** Similar to above comment, part of the intent was lost in the language. Text has been added for clarification. |

buckling due to the inherent eccentricity. While buckling is a common mode of failure in a wind turbine blade, it is driven predominately by the global structure and local geometry effects. Thus, even with the use of symmetric OP waves to reduce buckling during coupon testing, the reduction in material property in a compressed section of a blade is likely to have a more complex effect.  As such, no OP wave data is presented due for coupon test

[Figure]

**Figure 8: Peak Stress of OP and IP Waves at various fiber angles (left-to-right, respectively).**

Linear regression analysis demonstrateds that all the data displayed in Figure 8 fit best to exponentially decaying sinusoidal functions found by optimizing the coefficient of determination. This fit

10  has roots from strength of materials failure criteria where. Ffor an off-axis ply, the stresses rotate per:

$$\sigma'_{ij} = a_{ik} a_{jl} \sigma_{kl} \tag{7}$$

where $\sigma'_{ij}$ is the rotated local stress and $\sigma_{kl}$ is the global stress, and $a_{ik}$, $a_{jl}$ are direction cosines of the rotated region. With an interactive failure criterion, such as Tsai-Wu, the failure curve verses off axis angle is essentially a decaying stress rotation function which starts with  fiber dominated failure and quickly transitions to  matrix dominated failure with off-axis loading (Barbero, 2011).  Based on these results, this type of analysis can be used to

15  quickly assess the tension and compression failure strengths of wavy materials.

Due to variability in the specimen manufacturing processes, it was necessary to characterize the as-built flaw parameters prior to testing to ensure that all correlations  were performed accurately.  Through thickness, in-plane wave images for each layer were collected with the use of a Computer Tomography (CT) scanner where wave parameters were measured as displayed in Figure 6. For the case of IP waves,

20  each layer's off axis fiber angle was recorded and examples of the layer-by-layer variation in fiber angle is given in **Error! Reference source not found.**. Once testing of the IP wave samples was completed, the results were reviewed f or correlation to the maximum and average, through thickness wave angle. The analysis revealed very similar correlation traits, particularly when the maximum wave angle was considered making it the characteristic parameter

**Commented [u37]:** Fig 10 cmt 1: It is a shame the compression results for OP waves are not presented as these are probably the most interesting. Can it be included in spite of the variation? (REV #2 COMMENT -8) Fig 10 cmt 2: It is unclear how the sine-fit is generated, please elaborate. The fit originates from Eq (7), but the connection to Fig. 10 is not obvious. (REV #2 COMMENT -9) Fig 10 cmt 3: Can the plot be shown as a relative strength knockdown, ie. normalised by pristine strength?  (REV #2 COMMENT -10)

**Commented [n38R37]:** 8: The authors are not comfortable given the inability to scale the OP Wave in a representative manner. Thus, the data would be meaningless.  This is more directly explained.

9: Further explanation is given.

10: Plot reworked in this fashion.  Also simplified.

**Commented [n39]:** Figure 10. Use SI units. The fitting lines in (b) seem to be consisting of two lines (with transition points at about 10 and 13 degrees), and not made with a continuous mathematical function. (REV #1 COMMENT -14)

Figure 10. There are no compression data in (a)? (REV #1 COMMENT -15)

**Commented [n40R39]:** 14: Additional data points added to ensure consistency of trendline

15: The authors are not comfortable given the inability to scale the OP Wave in a representative manner. Thus, the data would be meaningless.  This is more directly explained.

**Commented [n41]:** change to "start with fibre dominated failure and transit to matrix dominated failure mode"? (REV #1 COMMENT -16)

**Commented [n42R41]:** We had this backwards.  We are saying that as the fiber angle increases from zero to 90 (longitudinal to transverse) the stress function goes from fiber to matrix. Thanks for catching!

used to describe as-built flaw magnitudes.  Out-of-plane waves were measured with the use of a high fidelity digital photographs and little migration during manufacturing was noted.

[Figure]

[Figure]

5    **Figure 9: Examples of layer-by-layer fiber wave variation. characterization**

**4.1.2 Porosity Trends**

Test specimens in the investigation have been manufactured using a vacuum bag technique, therefore it is necessary to include volume effects. A simple method for comparing results in this case is to normalize the failure stresses to
10   55% fiber volume ratio, $V_f$. Figure 10, left, shows a comparison between porosity content and the reduced strength. The void content was determined by image analysis of specimens from the same plates which were used for the test coupons. The void data are presented as a function of void content in the composite, to provide use to designers on that basis. Some discussion on the micromechanics of voids in the composite is warranted. The influence of voids on the mechanical properties has the effect of reducing the bulk modulus of the resin. While this does not have as great of an effect in tension, the reduced modulus has a significant effect for compression strength as the reduced modulus
15   does not support the fibers in compression as well as a stiffer matrix. While the results shown in Figure 10, left, are for an expected range, the entire dataset was compared with similar data from a prominent blade manufacturer with strong correlation between the two datasets (Figure 10, right) (TPI, 2010).  Based on these results, the BRC decided

**Commented [n43]:** Figure 11 shows a rather large variation in fibre angles between the 4 layers. This variation must somehow be included in the analysis as an uncertainty parameter. (REV #1 COMMENT -17)

**Commented [n44R43]:** The variation is not relevant because QC focus of the BRC.  Decision was made to not add another level in the uncertainty analysis.  In addition, data from a representative group is shown to clarify the variation due to manufacture.

to set 2% porosity as the upper threshold for acceptable porosity in blades. As such, further analysis was focused on this worst case, upper bound.

A vacuum bag technique was used to manufacture the test specimens in this investigation; therefore, it was necessary

5 to include volume effects. To accomplish this, failure stresses were normalized by the part thickness (i.e. normalize fiber volume for the same amount of resin and fiber) allowing for direct assessment of strength as impacted by porosity. As shown in **Error! Reference source not found.**, porosity has a greater adverse effect on compressive strength than tensile strength due to the reduction of resin performance.

[Figure]

[Figure]

[Figure]

10 **Figure 10: Reduction in Strength due to porosity in the 0 to 7% porosity range (left) and comparison with high porosity blade manufacturer dataset (right).**

Commented [n45]: At page 10, line 16, it is said that failure stress is normalised by the part thickness. It is not clear how this is done, it needs to be explained (REV #1 COMMENT -18)

Commented [n46R45]: Details were confused in the reduction of the work into this paper. Clarification has been made to this entire section.

Commented [u47]: Elaborate on why porosity has larger effect on compression than tension? (REV #2 COMMENT -11)

Commented [n48R47]: The entire porosity section has been reworked to better tell the story of the results pertinent to the rest of the work contained herein. As such, a narrower window of data has been assessed whereby the tension and compression trends are similar.

Commented [u49]: What is a typical level of porosity using a VARTM process? I imagine that >10% would indicate unwetted fibres, and can no longer be considered as porosity. Looks like the fit to compression data is quite vague, what is the correlation coefficient for the fits? Is there even a valid trend? What are the findings from other studies testing porosity? (REV #2 COMMENT -12)

Commented [n50R49]: Porosity figures have been updated as well [Figure 10] as noted in previous comment. Specifically, we identify that other than just % porosity, there are distribution and size considerations that are not considered herein, but could be part of future work.

Commented [n51]: Figure 12, It seems that both lines should start in 1. There are many samples with really high content of porosity, up to 14%. How is it ensured that the effect of porosity is not larger than the effect of the OP and IP waves? (REV #1 COMMENT -19)

Commented [n52R51]: Trends fixed to start at 1. The porosity section has been updated and refocused to more directly represent the results. Combined effects of defects were beyond the scope of this study and has been noted as future work.

**4.2 Comparison of Material Properties**

5    Material properties for each coupon were calculated and averaged for each flaw group and are shown in **Error! Reference source not found.** and **Error! Reference source not found.** with standard deviations  included for ultimate stress to indicate the distribution size of the coupons tested for each defect type.  To ensure accuracy of these values, comparisons were made between the control results and the results of similar tests published in the Montana State University Composite Material Database (**Error! Reference source not found.**).  M
10  odulus of elasticity and maximum strain were chosen as points of comparison as they were critical for analytical inputs and correlation.  These comparisons indicated that while the material properties compare acceptably for tension, the compressive failure strains for the test group appeared less accurate.  This was likely due to the unrestrained method of testing in compression which resulted in bending and buckling.  However, much less bending was noted in the flawed specimen, apart from OP waves.  The lack of bending prior to failure indicated that damage was occurring at
15  the flawed area prior to bending occurring suggesting that these data were acceptable.

**Table 3: Static properties for laminates tested in tension and calculated percentage of control laminates.**

| Tension | Control 0° | Control ±45° | 2% Porosity 0° | 2% Porosity ±45° | IP1 0° | IP1 ±45° | IP2 0° | IP2 ±45° | IP3 0° | IP3 ±45° | OP1 0° | OP1 ±45° | OP2 0° | OP2 ±45° | OP3 0° | OP3 ±45° |
|---|---|---|---|---|---|---|---|---|---|---|---|---|---|---|---|---|
| Ultimate Stress (MPa) | 990 | 112 | 950 | 103 | 344 | 109 | 226 | 107 | 521 | 108 | 417 | 84 | 742 | 101 | 752 | 102 |
| *Standard Dev* | (40) | (2.0) | (19) | (1.5) | (43) | (1.1) | (24) | (4.7) | (24) | (4.1) | (26) | (5.3) | (79) | (2.2) | (43) | (2.8) |
| % Control | -- | -- | 96% | 93% | 35% | 98% | 23% | 96% | 53% | 97% | 42% | 75% | 75% | 91% | 76% | 91% |
| Strain at Failure (%) | 2.64% | 2.61% | 2.54% | 3.32% | 1.66% | 3.07% | 1.66% | 2.41% | 1.66% | 3.23% | 4.77% | 4.91% | 4.92% | 4.06% | 4.56% | 4.43% |
| % Control | -- | -- | 96% | 127% | 63% | 118% | 63% | 92% | 63% | 124% | 181% | 188% | 186% | 156% | 173% | 170% |
| Modulus of Elasticity (GPa) | 41.1 | 16.2 | 39.6 | 16.6 | 34.8 | 16.8 | 24.1 | 16.6 | 39.6 | 18.7 | 17.3 | 5.9 | 30.8 | 16.1 | 31.2 | 15.3 |
| % Control | -- | -- | 96% | 103% | 85% | 104% | 59% | 100% | 96% | 115% | 42% | 36% | 75% | 100% | 76% | 94% |
| Poisson's ratio | 0.27 | | -- | | -- | | -- | | -- | | -- | | -- | | -- | |

| Tension | Control 0° | Control ±45° | Porosity 0° | Porosity ±45° | IP 1 0° | IP 1 ±45° | IP 2 0° | IP 2 ±45° | IP 3 0° | IP 3 ±45° | OP 1 0° | OP 1 ±45° | OP 2A 0° | OP 2A ±45° | OP 4A 0° | OP 4A ±45° |
|---|---|---|---|---|---|---|---|---|---|---|---|---|---|---|---|---|
| Ultimate Stress (MPa) | 990 | 112 | 950 | 103 | 521 | 108 | 344 | 109 | 226 | 107 | 417 | 84 | 742 | 101 | 752 | 102 |
| *Standard Dev* | (40) | (2.0) | (19) | (1.5) | (24) | (4.1) | (43) | (1.1) | (24) | (4.7) | (26) | (5.3) | (79) | (2.2) | (43) | (2.8) |
| % Control | -- | -- | 96% | 93% | 53% | 97% | 35% | 98% | 23% | 96% | 42% | 75% | 75% | 91% | 76% | 91% |
| Strain at Failure (%) | 2.64% | 2.61% | 2.54% | 3.32% | 1.66% | 3.23% | 1.66% | 3.07% | 1.66% | 2.41% | 4.77% | 4.91% | 4.92% | 4.06% | 4.56% | 4.43% |
| % Control | -- | -- | 96% | 127% | 63% | 124% | 63% | 118% | 63% | 92% | 181% | 188% | 186% | 156% | 173% | 170% |
| Modulus of Elasticity (GPa) | 41.1 | 16.2 | 39.6 | 16.6 | 39.6 | 18.7 | 34.8 | 16.8 | 24.1 | 16.6 | 17.3 | 5.9 | 30.8 | 16.1 | 31.2 | 15.3 |
| % Control | -- | -- | 96% | 103% | 96% | 115% | 85% | 104% | 59% | 100% | 42% | 36% | 75% | 100% | 76% | 94% |
| Poisson's ratio | 0.27 | | -- | | -- | | -- | | -- | | -- | | -- | | -- | |

20  **Table 4: Static properties for laminates tested in compression and calculated percentage of control laminates.**

**Commented [n53]:** Page 11, line 5, it is said that model predictions of load displacement curves were accurate to within +/-5%; where is that shown?
(REV #1 COMMENT -20)

**Commented [n54R53]:** Misplaced sentence from previous editing.

**Commented [n55]:** Page 11, line 9, "Standard deviations shown for ultimate stress indicate the consistency of each test", what is meant by that
(REV #1 COMMENT -21)

**Commented [n56R55]:** Clarification text has been added

Marked Up Manuscript Version

[revised manuscript text omitted]

**Commented [u61]:** The content of this section is relative thin. Can this section be elaborated and show some DIC strain plots of the various strain components for instance? (REV #2 COMMENT -13)

**Commented [n62R61]:** Figure reconstructed as recommended and text changed/modified to better show and explain damage progression

[Figure]

**Commented [u63]:** Relate the damage stages to the stress-strain curve or at least state the load fraction at the given spot. The DIC plot to the far right is useless; no scalebar is shown and it does not state which strain component is shown. (REV #2 COMMENT -14)

**Commented [n64R63]:** Figure reconstructed as recommended and text changed/modified to better show and explain damage progression

[Figure]

**Figure 11:**  Stress-strain response of IP3 wave coupon shown at increasing average full field strain with associated DIC strain fields identifying damage progression.

In summary, it should be noted that the IP3  case had decreases in material strength, and significant degradation was noted, making the result that this case was optimal for baseline use for modelling efforts (Riddle et al, 2017; Nelson et al, 2017). Further, it was decided that since this case had a fiber misalignment angle close to 30°, it would be a good median case for these endeavors. The resulting stress-strain curves, utilizing the DIC data from this test group, for this IP wave case in tension and compression are found in **Error! Reference source not found.**. Data beyond failure and maximum stress was gathered to begin to establish a comprehensive understanding of the material to be applied to future work with larger substructures and structures. As such, this geometry and these results were utilized as the baseline model for experimental/analytical correlation of each modelling type outlined below.

[Figure]

**Figure 12:  Stress-strain of IP Wave 1 in tension and compression utilized for baseline model correlations with associated experimental variability.**

**4.2.2 OP Waves Analysis**

Test results for the OP wave groups are also noted in **Error! Reference source not found.** and **Error! Reference source not found.** with a representative stress-strain response shown in **Error! Reference source not found.**.  Results from test observation and the DIC suggest that each of the OP wave groups was noted to have similar damage progression in tension compared to IP waves where delamination took the place of the matrix damage between the fiber tows.  A strain levels increased, cracks initiated in the resin between the layers at the ends of the wave before delaminating.  However, unlike the behavior of IP waves, after delamination and significant fiber straightening, the failure area for the OP wave specimens was concentrated at the peak area of the wave.  This was due to the fibers being pulled straight and the center of bending being at the peak of the wave.  Compression testing of the OP waves proved to be very difficult, as large wavelengths necessitated a long unsupported gage length resulted in significant bending and ultimately  As such, significant decreases in calculated moduli of elasticity, ultimate strength, and strain at failure were noted and results were considered unusable for correlation given these responses.

**Commented [u65]:** For OP waves it is unclear what the failure mode is; is it interlaminar delamination or buckling? What is the failure mode in compression? (REV #2 COMMENT -15)

**Commented [n66R65]:** Information added and reorganized to more clearly identify damage progression in both tension & compression

[Figure]

**Figure 13: Stress-strain of OP Wave 34A in tension utilized for initial OP wave model correlations.**

It must be noted that the wave forms for all the OP1 group delaminated during testing. This resulted in an extreme decrease in the ultimate stress and stiffness results of the OP1 groups in both tension and compression. As such, OP1
5      was deemed unusable for correlation in both tension and compression.

The two other cases, OP2 2A and OP4 4A, were found to have a more consistent response. Ultimate stress and strain at failure values for the OP 2A and OP3 4A 0° and ±45° tension groups were decreased compared to the control but were increased compared to the IP waves. Thus, moduli of elasticity values were similar to the control due to load being transferred more consistently through the wave than seen with IP waves due to the configuration described
10     above. Given the consistency of these waves in tension, the OP3 4A case was utilized for correlation. ~~However, compression testing of the OP waves proved to be very difficult, as large wavelengths necessitated a long unsupported gage length. This resulted in significant bending as the load transferred through the wave. As such, significant decreases in calculated moduli of elasticity, ultimate strength, and strain at failure were noted and results were considered unusable for correlation given these responses.~~ Overall, the static testing performed allowed for initial
15     analysis while determining convergence points for analytical models.

**5 Conclusions and Future Work**

Using this consistent framework that was established and validated, defects common to wind turbine blades have been quantified. To effectively characterize, categorize, and analyze defects, the frame requires accurate data collection following consistent scientific procedures. With proper characterization, it is possible to establish the mechanical
20     response of a flaw using laboratory testing. Results from static testing indicate that there is a strong correlation between flaw parameters and mechanical response. Since the flaws went across the entire width of the sample, applying these knockdowns directly is conservative, but may not be realistic, especially if surrounding material in a blade structure can redistribute loads from local failures. Going forward, the characterization techniques described herein may be applied to incoming data will enable the generation of a statistically significant and comprehensive flaw database.

This work provides a sound starting point, but only constitutes the building blocks for a comprehensive reliability program aimed at reducing failures as a result of defects. Since reliability estimation is inadequate for composite structures due to the uncertainties, a probabilistic approach is required to achieve an acceptable level of confidence. This approach must consider multi-scale mechanical property variability, damage/defect detection, damage

5    progression, residual strength analysis, global, and macro structural response

[revised manuscript text omitted]